# Epitaxial nucleation and lateral growth of high-crystalline black phosphorus films on silicon

Yijun Xu[1,2,7], Xinyao Shi[1,7], Yushuang Zhang[3,7], Hongtao Zhang[4], Qinglin Zhang[3], Zengli Huang[5], Xiangfan Xu [6], Jie Guo[6], Han Zhang [2✉], Litao Sun [4], Zhongming Zeng[1], Anlian Pan[3✉] & Kai Zhang [1✉]

Black phosphorus (BP) is a promising two-dimensional layered semiconductor material for next-generation electronics and optoelectronics, with a thickness-dependent tunable direct bandgap and high carrier mobility. Though great research advantages have been achieved on BP, lateral synthesis of high quality BP films still remains a great challenge. Here, we report the direct growth of large-scale crystalline BP films on insulating silicon substrates by a gas-phase growth strategy with an epitaxial nucleation design and a further lateral growth control. The optimized lateral size of the achieved BP films can reach up to millimeters, with the ability to modulate thickness from a few to hundreds of nanometers. The as-grown BP films exhibit excellent electrical properties, with a field-effect and Hall mobility of over 1200 cm$^2$V$^{-1}$s$^{-1}$ and 1400 cm$^2$V$^{-1}$s$^{-1}$ at room temperature, respectively, comparable to those exfoliated from BP bulk crystals. Our work opens the door for broad applications with BP in scalable electronic and optoelectronic devices.

[1] CAS Key Laboratory of Nano-Bio Interface & Key Laboratory of Nanodevices and Applications, i-Lab, Suzhou Institute of Nano-Tech and Nano-Bionics (SINANO), Chinese Academy of Sciences, Suzhou 215123, China. [2] Shenzhen Engineering Laboratory of phosphorene and Optoelectronics, Collaborative Innovation Center for Optoelectronic Science and Technology and Key Laboratory of Optoelectronic Devices and Systems of Ministry of Education and Guangdong Province, Shenzhen University, Shenzhen 518060, China. [3] Key Laboratory for Micro-Nano Physics and Technology of Hunan Province, College of Materials Science and Engineering, School of Physics and Electronic Science, Hunan University, Changsha 410082, China. [4] SEU-FEI Nano-Pico Center, Key Lab of MEMS of Ministry of Education, Southeast University, Nanjing 210096, China. [5] Vacuum interconnected Nanotech Workstation (Nano-X), Suzhou Institute of Nano-Tech and Nano-Bionics (SINANO), Chinese Academy of Sciences, Suzhou 215123, China. [6] Center for Phononics and Thermal Energy Science, School of Physics Science and Engineering, Tongji University, Shanghai 200092, China. [7] These authors contributed equally: Yijun Xu, Xinyao Shi, Yushuang Zhang. ✉email: hzhang@szu.edu.cn; anlian.pan@hnu.edu.cn; kzhang2015@sinano.ac.cn

Black phosphorus (BP) is a promising two-dimensional (2D) layered semiconductor for next-generation electronics and optoelectronics, which owns high carrier mobility, thickness-dependent direct band gaps from 0.3 eV to ~ 1.5 eV as well as anisotropic nature[1–12]. To explore the full potential of BP, especially for various devices that can be integrated in silicon flows, it basically requires reliable synthesis of high-quality large-area BP films. Despite considerable efforts and some successful examples to date[13–15], lateral growth of high-crystalline BP films, which hold promising electrical properties and meet the silicon device-scaling requirements, has not been realized. Here, for the first time, we report the direct growth of large-scale crystalline BP films on insulating silicon substrates by a newly developed gas-phase growth strategy with an epitaxial nucleation design and a further lateral growth control. The optimized lateral size of the achieved BP films can reach up to millimeters, with the thickness be modulated from few to hundreds of nanometers. The as-grown BP films exhibit excellent electrical properties with field-effect and Hall mobility of over $1000 \, \mathrm{cm^2 \, V^{-1} \, s^{-1}}$ and $1400 \, \mathrm{cm^2 \, V^{-1} \, s^{-1}}$ at room temperature, respectively, and current on/off ratio up to $10^6$, comparable to those exfoliated from BP bulk crystals. In addition, extraordinary lamellar structure is explored with the synthesized BP films, which brings superior infrared optical characteristics with a strengthened infrared absorption and photoluminescence (PL). Our work demonstrates an important step forwards large-scale preparation of crystallized BP films and open the door for broad applications in scalable optoelectronic devices and compact integrated circuits.

Traditionally, BP can be prepared through the phase transfer of white or red phosphorus (RP) by high pressure and high-temperature route, mercury catalysis or by re-crystallization from a Bi-flux[16–18]. Making use of the phase change, recent progress on growing BP by a mineralizer-assisted gas-phase transformation method further improve the yield and crystallinity[1,19–24]. However, to date, only BP in bulk crystals have been achieved in this way, whereas with no report on the success of growing crystalline BP thin films directly on wafers[25]. Quite some efforts have been devoted to develop the direct growth of BP films on substrates recently. Lau et al. first report the growth of 2 nm-thick BP film on silicon using a pulsed laser deposition. However, the obtained film is amorphous, and the fabricated devices exhibit very low electronic performance[13]. Xia's group synthesized a large-area (up to 4 mm) thin BP film on a flexible polyester substrate by converting RP film into BP film by pressurization in an anvil cell. But the grain size of the grown BP film is only 10 nm, bringing a very low mobility of only $\sim 0.5 \, \mathrm{cm^2 \, V^{-1} \, s^{-1}}$, far from the practical applications[14]. Further progress was made recently to improve the crystallinity of the BP films by a similar pressurization approach in transforming RP to BP at 1.5 GPa and 700 °C on sapphire substrates. The mobility of the as-grown BP films can be improved to be $\sim 160 \, \mathrm{cm^2 \, V^{-1} \, s^{-1}}$ [15]. However, the extremely high-pressure might be a disadvantage to the choice of the growth substrates for BP. Thus, a milder growth conditions for high-quality BP films becomes necessary.

## Results

**Nucleation and growth of BP crystalline nanosheets.** In this work, we develop a novel growth strategy in designing the nucleation of BP on wafers with the assistance of polyphosphide $Au_3SnP_7$ as the nucleation seeds. Based on the effective nucleation and spatial control, we realized, for the first time, large-size growth of highly crystalline BP films on insulating substrates like silicon. $Au_3SnP_7$ behaves an important intermediate product when utilizing Au or AuSn as the precursor to grow BP in the previously reported mineralizer-assisted gas-phase transformation method. It

is worth noting that, $Au_3SnP_7$ is very stable during the BP growth and its (010) plane has atomic lattice match with the (100) plane of BP. We propose to exploit this feature to control the nucleation and growth of BP by strategically forming $Au_3SnP_7$ on substrates (see Fig. 1a). In realizing it, $Si/SiO_2$ substrate with intentionally deposited thin Au film was enclosed in evacuated silica glass tubes together with excessive red P and Sn to form distributed $Au_3SnP_7$ on silicon during the heating process at 750 °C. The formation of $Au_3SnP_7$ can be described by the following equations[19]:

$$Au_2P_{3(s)} + AuSn_{(s)} + 4P_{(s,red)} = Au_3SnP_{7(s)} \qquad (1)$$

$$11/8Au_2P_{3(s)} + 2/8AuSn_{4(s)} + 23/8P_{(s,red)} = Au_3SnP_{7(s)} \qquad (2)$$

$$5/4Au_2P_{3(s)} + 2/4AuSn_{2(s)} + 13/4P_{(s,red)}\big) = Au_3SnP_{7(s)} \qquad (3)$$

$$9/6Au_2P_{3(s)} + 2/6Sn_3P_{4(s)} + 7/6P_{(s,red)}\big) = Au_3SnP_{7(s)} \qquad (4)$$

$$6/4Au_2P_{3(s)} + 1/4Sn_4P_{3(s)} + 7/4P_{(s,red)} = Au_3SnP_{7(s)} \qquad (5)$$

$SnI_4$ is added as the mineralizer to promote the reaction. The nucleation and growth of BP films on $Si/SiO_2/Au_3SnP_7$ occur when holding at lower temperature of ~ 500 °C and the further cooling stage (see more growth details in the Method and Supplementary Fig. 1), and probably evolve according to the scenario as following: when the temperature decreased to ~500 °C, $P_4$ vapor condensed and precipitated on the surface of $Au_3SnP_7$. As a fact, the formed $Au_3SnP_7$ is in a regularly shaped crystal, usually with a lateral size of hundreds of nanometers that can be seen in the Fig. 1b. The element mapping of the $Au_3SnP_7$ (Supplementary Fig. 2) proves the homogeneous distribution of Au, Sn, and P elements. The high crystallinity of the $Au_3SnP_7$ was confirmed from the sharp and intense diffraction spots in the selected area electron diffraction (SAED) pattern (Supplementary Fig. 3). The lattice spacing of the $Au_3SnP_7$ is 3.2 Å along (010) direction and 5.4 Å for (110), which is similar to the lattice parameters of BP (Fig. 1b, insert, and Supplementary Fig. 4). These characteristics make $Au_3SnP_7$ served as a proper nucleation seed for the epitaxial growth of BP. As the arrangement of six-membered P ring fragments of BP along the (100) plane and $Au_3SnP_7$ along the (010) plane own very similar distances between each other (as depicted in Supplementary Fig. 4), it kinetically prefers the route with $P_4$ phase transformed to be BP and epitaxially nucleated on $Au_3SnP_7$. This assumption can be firmly proved by the high-magnification cross-sectional transmission electron microscopy (TEM) images of the nuclei formed on substrate, where the ordered coexistence of BP and $Au_3SnP_7$ and the atomically sharp interface between them are clearly displayed (Fig. 1c). A transition state, as the so-called small BP nanosheets over the nuclei center, is observed following the epitaxial nucleation stage. With further heating process, the small BP nanosheets with thickness ~ 5 nm thermodynamically grow and fuse across with each other to be large planar ones (as depicted in Fig. 1a with proof images shown in Fig. 1d). Finally, large-area BP thin films grown on silicon substrates (schematically shown Fig. 1a) may be achieved after cooling down at a slow cooling rate of 1 °C min$^{-1}$.

Figure 1e presents a typical BP film with a lateral size over 200 μm grown directly on the $Si/SiO_2$ substrate. The BP film exhibits a clean and flat surface with a thickness ~ 150 nm (Supplementary Fig. 5a). In fact, the BP film can be as large as millimeters in lateral size with a thicker morphology (Supplementary Fig. 6). The strong peaks at (0 2n 0) examined in the X-ray powder diffraction (XRD) measurements (Fig. 1f) demonstrate a clearly layered structure of the as-grown BP films. The XRD results

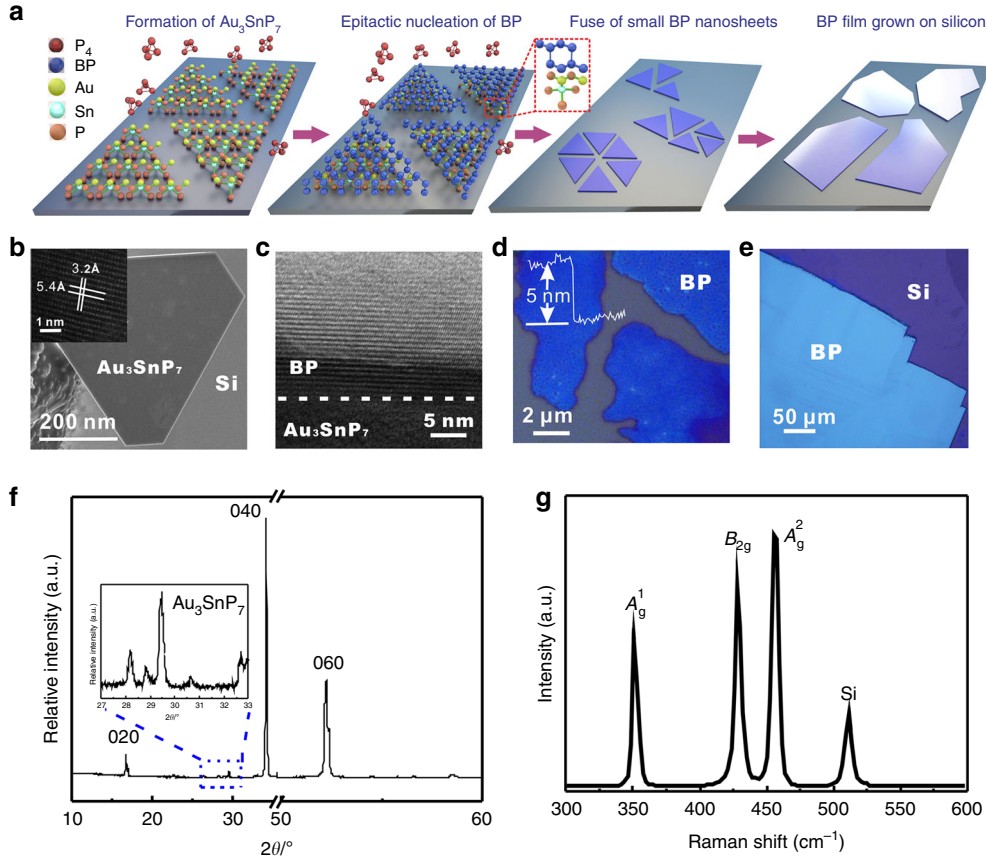

**Fig. 1 Nucleation mechanism and growth process of BP crystalline films. a** Schematic illustration showing the formation of $Au_3SnP_7$, epitactic nucleation of BP, fuse of small BP nanosheets into large ones on substrate. **b** SEM of a $Au_3SnP_7$ nucleation seed grown on $Si/SiO_2$ substrate. Inset shows the HRTEM image of the obtained $Au_3SnP_7$. **c** Cross-sectional HRTEM image of a layered crystalline BP nanosheet grown on $Au_3SnP_7$. **d** Optical imaging of the transition state during the growth. Fusing of small BP nanosheets to be large planar ones is clearly displayed. Line-scan profile indicates these BP nanosheets during the transition states are with thickness of ~ 5 nm. **e** Optical image of a representative BP film grown on $Si/SiO_2$ substrate with a lateral size over 200 μm. **f** XRD spectra of the synthesized BP film. A typical orthorhombic structure of BP is evidenced. Extra peaks from $Au_3SnP_7$ as the nucleation buffer layer between BP and silica are also examined. **g** Raman spectrum of the BP film grown on $Si/SiO_2$ substrate, which prove its high crystallinity and purity.

indicate a typical orthorhombic structure of our synthesized BP film with lattice constants of $a = 3.32$ Å, $b = 4.38$ Å, and $c = 10.48$ Å, consistent with single crystal BP[26]. In addition, some extra peaks between 27 and 33° besides those belonging to BP were examined, which are expected to be attributed to $Au_3SnP_7$[27]. This also provides a strong evidence of the existence of $Au_3SnP_7$ as the nucleation seeds enabling the direct growth of BP film on silicon substrates as we proposed. The experimental proof with $Au_3SnP_7$ nano-islands beneath the as-grown BP film further confirms the epitactic growth of BP films above $Au_3SnP_7$ nucleation seeds (Supplementary Fig. 7). The formation of $Au_3SnP_7$ located only at the bottom end of the BP films guarantees the high quality and purity of the BP samples, which was proved by the Raman and X-ray photoelectron spectroscopy (XPS) characterizations. Raman spectra show characteristic peaks of BP located at 364, 438, and 465 cm$^{-1}$ (Fig. 1g), corresponding to $A^1_g$, $B_{2g}$, and $A^2_g$ vibration modes, respectively[2]. And, during the XPS scanning, the as-grown BP films exhibit the $2p_{3/2}$ and $2p_{1/2}$ doublet at 130.15 and 130.9 eV (Supplementary Fig. 5b), respectively, that is the characteristics of crystalline BP.

Excessive and rapid transportation of $P_4$ vapor during the growth hinders the formation of BP thin films but rather with thick flakes. For a fine control, several growth routes for reducing the $P_4$ source involved in the transformation into BP were employed. First, the RP was placed at the low temperature (LT)

side while evaporated into $P_4$ molecules and transported thermodynamically towards high temperature (HT) side, where BP films nucleate and grow at the far end of the substrate. The assignment of the transportation of $P_4$ molecules from LT to HT may reduce the amount of $P_4$ source involved in the growth. Furthermore, massive $Si/SiO_2$ substrates coated with Au films were vertically stacked and placed in the evacuated silica tube. The limited space among the substrates confines the diffusion of $P_4$ molecules on the top of the $Au_3SnP_7$ nucleation seeds, which further limits the source amount. The experimental setup of the BP growth is schematically shown in Fig. 2a. With a proper control of the temperature gradient between the source at LT and substrates at HT, the thickness as well as the lateral size of the BP film can be well under controlled (Fig. 2b–i). BP films as thin as down to several nanometers were directly grown on $Si/SiO_2$ substrates, as shown with the optical image in Fig. 2b and corresponding AFM profile in Fig. 2f. Usually, the thicker of the film, the larger of the size may be achieved during the growth. Several hundred of microns to sub-millimeter size BP films can be easily grown when the thickness is around or over 100 nm.

**Microstructure of the BP films**. The microstructure of the synthesized BP films was systematically studied with high-resolution microscopy. Figure 3a shows the cross-sectional transmission

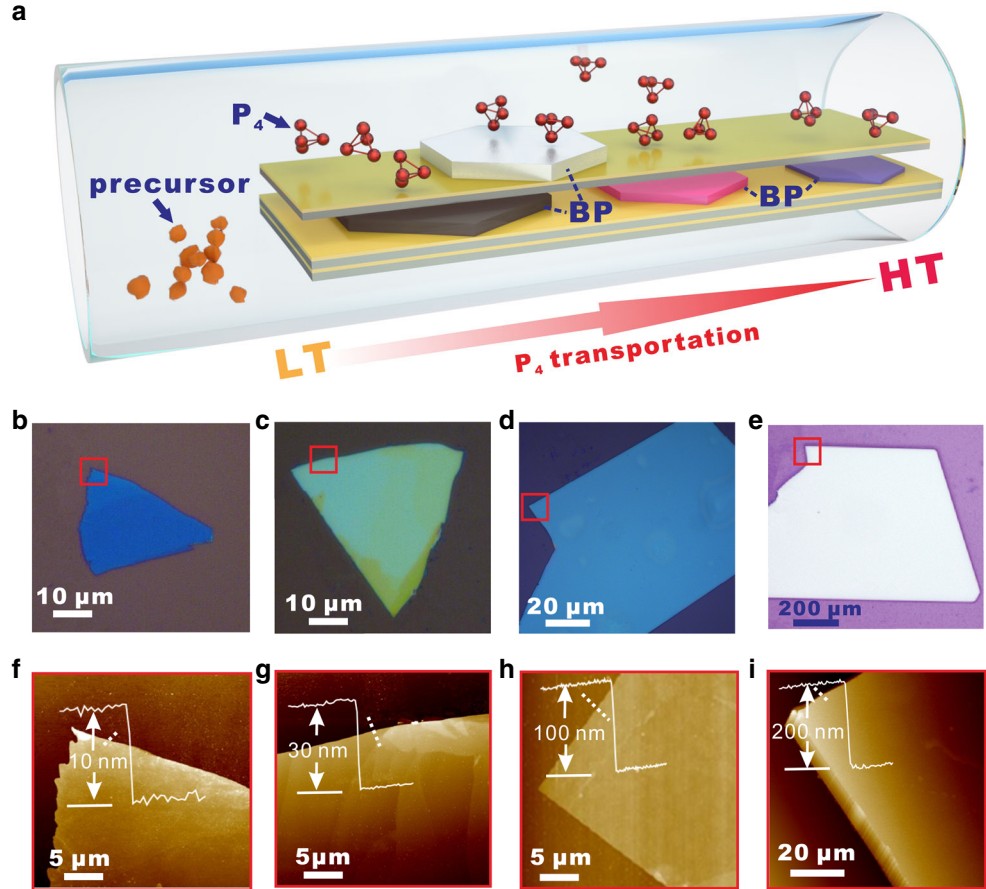

**Fig. 2 Control of the lateral size and thickness of the BP films during growth. a** Schematic view of the experimental setup for the BP growth. Specially, the RP was evaporated to $P_4$ and diffused from low temperature (LT) side toward high temperature (HT) side. And, a massive of Au/Si substrates were vertically stacked and placed in the evacuated silica tube. These routes were employed for reducing the $P_4$ source involved in the transformation into BP. **b–e** BP films with lateral size ranging from several micronmeter to sub-millimeter under a proper control of the temperature gradient between the source at LT and substrates at HT. **f–i** The corresponding AFM images indicating the thicknesses of the BP films, ranging from several nanometers to hundred of nanometers.

electron microscopy (TEM) image of the as-grown BP film on silicon substrate, where the sharp interfaces between $Si/SiO_2$, $Au_3SnP_7$ nucleation seed layer and BP film can be clearly observed. The corresponding energy-dispersive X-ray spectroscopy (EDS) element mapping images clearly display the configuration of BP film directly grown on silicon substrate with a thin $Au_3SnP_7$ layer as the nucleation buffer. It also indicates the purity of the as-grown BP film without any atomic diffusion and contamination during the designed growth route. According to the cross-sectional SEM image, an obvious layered structure of BP (~200 nm thick) on $Au_3SnP_7$/Silicon can be observed (Fig. 3b). Interestingly, our synthesized BP films are found to own extraordinarily lamellar structure. Supplementary Fig. 8a shows a representative high-magnification cross-sectional SEM image of the as-grown BP film, which is constructed by amount of alternating BP atomic layer units and minor gaps. If examining through the BP films with the high-magnification TEM, we can also find the fact that the BP film is regularly composed with few nanometer-thick (~5–10 nm) BP atomic layers as unit cells, which are separated with each other by tiny gaps (Fig. 3c). This corresponds well with the cross-sectional SEM imaging results. The minor gaps are probably formed when the layer units sliding across each other during the covalence of adjacent BP nanosheets. The BP nanosheets tend to loosely stack along the z-direction rather than fusing into crystals. It is quite different from those of the conventional BP bulk crystals synthesized before, which

behaves a much densely squeezed layered (DSL) structure (Supplementary Fig. 8b). Figure 3d gives the cross-sectional image of the edge of the BP atomic layer derived from the as-grown BP film, which displays a clear layered structure with 0.5 nm spacing, consistent with the characteristic of crystalline BP. The atomic structure of the crystalline BP film is shown by the high-resolution TEM (HRTEM) image in Fig. 3e, where clear phosphorus atoms without visible impurities or defects indicate perfect orthogonally symmetric structure. The measured lattice spacing of 3.2 Å in the (100) direction and 4.2 Å in the horizontal direction for the (001) lattice matches well with the lattice orientation along zigzag and armchair directions, respectively, as demonstrated by the inset schematic model. AB stacking characteristics of the BP atomic layers is also present in the HRTEM image. The corresponding SAED pattern (shown in insert of Fig. 3e, as well as Supplementary Fig. 9) suggests the high crystallinity nature of the BP film reflected from the sharp and intense diffraction spots in the SAED pattern. The SAED pattern can be well indexed to an orthorhombic structure and the zone axis was determined as the [010] direction, which agrees with our XRD data and previous reports on exfoliated crystalline BP.

**Electrical properties of the BP films**. To evaluate the electrical performances of the synthesized BP films, we measured their field-effect transport properties. A series of back-gated field-

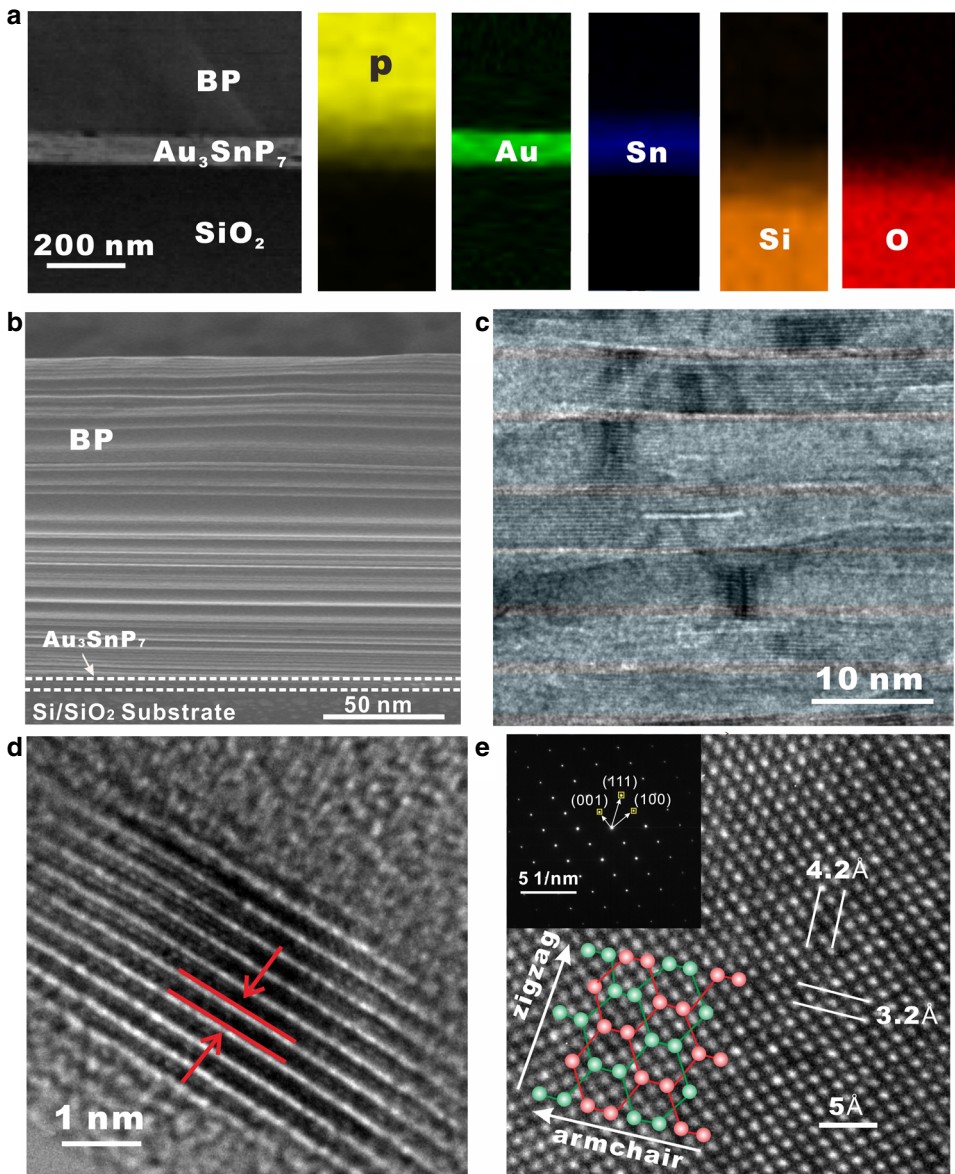

**Fig. 3 Microstructure of the BP films grown on silicon. a** Cross-sectional TEM image of a BP film grown on silicon substrate and the corresponding element mapping of the P, Au, Sn, Si and O. **b**, **c** The SEM and close-up HRTEM image of the BP film in **a**. It uncovers that each BP film is regularly composed with few nanometer (~ 5–10 nm)-thick BP atomic layers as unit cells, with tiny gaps between each other. **d** Cross-sectional HRTEM image of a BP atomic layer derived from the as-grown BP film, displaying a clear layered structure with 0.5 nm spacing. **e** HRTEM image of the BP film, representing a defect-free atomic structure. The lattice parameters match well with those of bulk BP crystals, showing an orthorhombic symmetry in AB stacking mode. Insert displays the corresponding SAED pattern of the BP film, demonstrating its single-crystalline characteristic with zone axis along [100] direction.

effect transistors (FETs) were fabricated directly using the as-grown BP films (usually the ones in thickness of ~ 8 nm were chosen) on highly doped silicon substrates with 285 nm $SiO_2$ as gate dielectrics and Cr/Au as source and drain electrodes. The source-drain I–V is linear, indicates that the Schottky barrier at the electrode-BP interface is low (Fig. 4a, insert). The source-drain current $I_{ds}$ vs. gate voltage $V_g$ at fixed bias voltage of $V_{ds} = 0.05$ V (Fig. 4a), exhibits a typical p-type ambipolar transport behavior with the threshold gate voltage located ~30 V. The extracted field-effect mobility shows an average of 1200 $cm^2\,V^{-1}\,s^{-1}$ at room temperature for the multiple devices (Fig. 4b, black line, Supplementary Note 1), which is comparable to the reported experimental values for mechanically exfoliated BP from bulk crystals[1]. We also examine the temperature dependence of the carrier mobility, which behaves

relatively stable at the range of 10 K to 300 K. The BP FET also displays an excellent current on/off ratio as high as $10^6$ for gate voltages in the range of −40 to 40 V (Fig. 4b, red line), comparable to previous measurements. To further evaluate the carrier characteristics of the synthesized BP films, standard Hall-bar devices were fabricated and measured at different temperatures varying from 1.5 K to 300 K (Fig. 4c). The hole Hall mobility saturates when T < 10 K and reaches up to 2200 $cm^2\,V^{-1}\,s^{-1}$ (the calculation of the Hall mobility can be seen in Supplementary Note 1). And, the hole concentration reaches maximum of $6.25 \times 10^{12}\,cm^{-2}$ at 300 K, which is comparable to the former report. The superior carrier mobility together with the high on/off current ratio characteristics further proves the high quality of the synthesized BP films and promises the key metrics for high-speed logic devices.

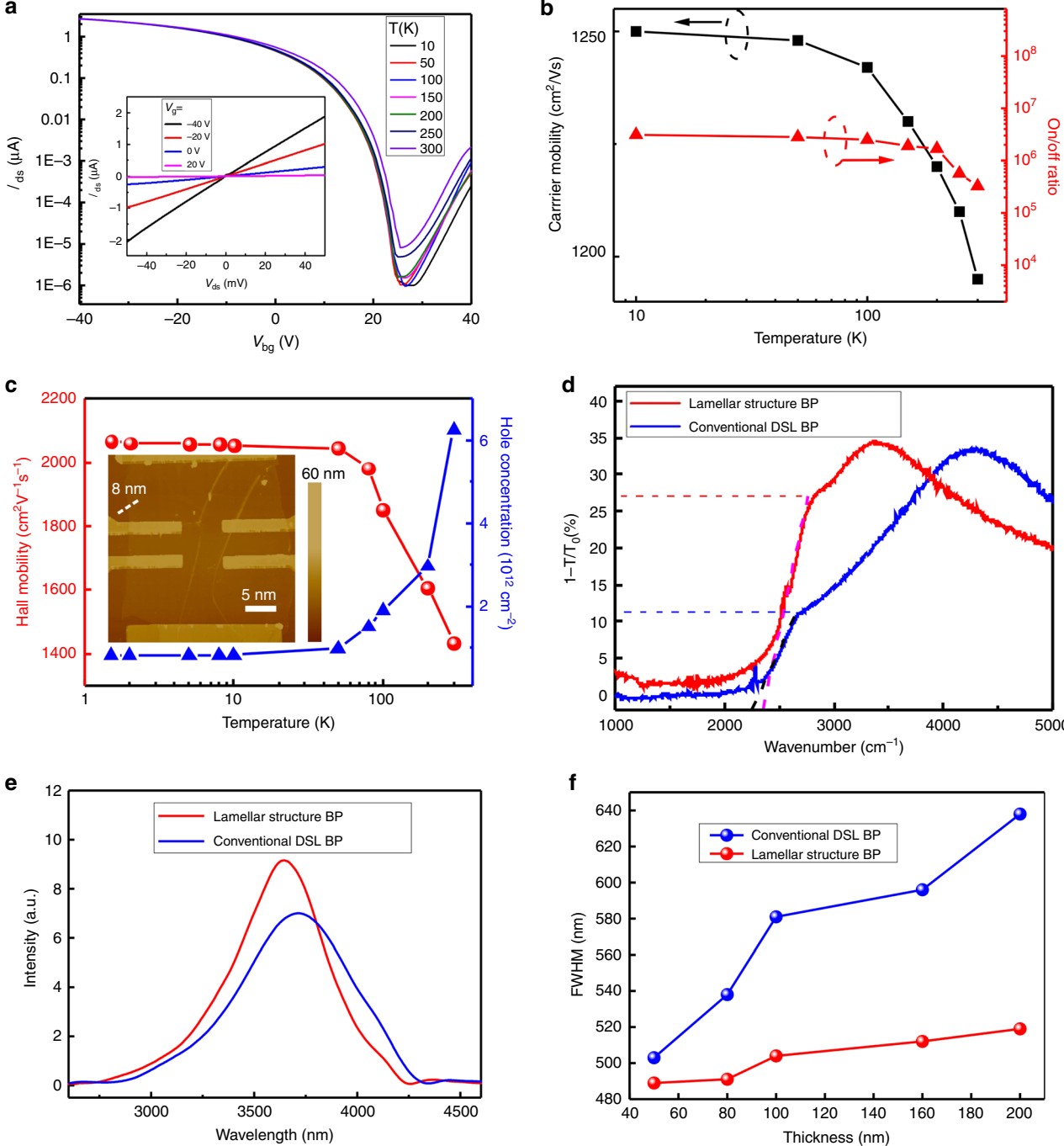

**Fig. 4 Electrical and infrared optical properties of the as-grown BP films. a** Output and transfer curves of the FETs utilizing the synthesized BP film as channel materials. Transfer characteristics was measured under gate voltage $V_g$ swept from −30 to 50 V with fixed source-drain voltage $V_{ds} = 100$ mV. The output result is demonstrated in the insert, measured under $V_{ds}$ swept from −50 to 50 mV, and different gate voltages from −40 to 20 V with 20 V as step. Electrical characterizations demonstrate a typical p-type ambipolar transport behavior of the BP sample. **b** Carrier mobility and current on/off ratio as a function of temperature in a synthesized BP film. Field-effect mobility of over 1200 cm$^2$ V$^{-1}$ s$^{-1}$ and excellent current on/off ratio as high as 10$^6$ for gate voltages in the range of −40 to 40 V can be achieved. **c** Hall mobility (red line) and carrier concentration (blue line) of the BP film as a function of temperature. Insert is the AFM topography image of a Hall-bar device employed for the characterization, where the line profile indicates the thickness of the BP is ~8 nm. **d** Infrared absorption spectra of the as-grown lamellar structure BP film (red line) and the BP derived from conventional bulk crystal (blue line). **e** PL spectra of the as-grown lamellar structure BP film (red line) compared with BP derived from conventional bulk crystal (blue line). **f** The thickness dependence of the full width at half maximum of the PL spectra for the as-grown lamellar structure BP film (red line) and those derived from conventional BP bulk crystal (blue line). All the spectra measurements were carried out at ambient conditions, and employing two types of BP with the same thicknesses.

**Infrared optical properties of the as-grown BP films**. The optical properties of the as-grown BP films have also been investigated at the infrared regime. Figure 4d shows the infrared absorption spectra of as-grown lamellar structure BP films and that derived from conventional DSL BP. Both the two types of BP show an absorption edge around 2400 cm$^{-1}$, indicating a same optical band gap of 0.3 eV. Compared with the conventional DSL BP, the as-grown lamellar structure BP films display a much stronger absorbance, with 27% relative to 11% when measured at the same thickness of 120 nm. Consistent results were also acquired in other samples with various thicknesses (Supplementary Fig. 10). The strengthened infrared absorption is probably attributed to the constructive interference from the regular nano-gaps among BP atomic layer units within the lamellar microstructure[28]. Infrared PL characterizations were also systematically carried out. A clear stiffening of the PL peak was observed when comparing the PL spectra of lamellar structure BP films with that derived from conventional DSL BP, with the same thickness of 200 nm (Fig. 4e). We ascribe it to the unique lamellar structure of the as-grown BP films, where the nano-gaps among distributed BP atomic layers decrease the charge transfer between the layers and hence lead to a more-efficient confinement of electron-hole pairs in each layer. This results in the enhancement of electron-hole recombination thus the increase of luminescence. Similar performance can be also explored with different BP samples with various thicknesses (Supplementary Fig. 11a, b). Usually, with the increasing of the thickness, the accumulated defects in BP will lead to the broadening of the PL peaks[29,30]. While, the confinement of electron-hole pairs in lamellar structure BP could somehow weaken the defect effect and hence maintain the full width at half maximum of PL peaks almost unchanged, as shown in Fig. 4f. Supplementary Fig. 11d displays the evolution of the PL spectra of the BP film as a function of the laser power. The peak intensity increase with the increasing of the excitation power, and did not decrease even at the high power of 48 mW, suggesting the stability of the BP film. The anisotropic optical properties of the BP were examined by angle-resolved polarized Raman spectroscopy. Both the as-grown lamellar structure BP films and conventional DSL BP exhibit the similar anisotropic optical properties, as confirmed in Supplementary Fig. 12. These optimal PL features with the as-grown BP films are promising for the applications in infrared optical sources, like micro-LED, lasers, etc. Excellent infrared photodetection performance with the synthesized BP films is also demonstrated (Supplementary Figs. 13, 14, calculation of photoresponsivity and photoconductive gain was shown in Supplementary Note 2).

## Discussion

In conclusion, a novel growth strategy in designing the nucleation and further spatial growth of large-area high-crystalline BP films on silicon was demonstrated. $Au_3SnP_7$ formed on silicon plays as a key role for the epitaxial nucleation and growth of BP. The thickness of the BP films can be well controlled from several nanometers to hundreds of nanometers. The synthesized BP films own an extraordinarily lamellar microstructure, composed by few nanometer-thick ($\sim 5-10$ nm) BP layers as unit cells and weak bonding among unit laminated layers. The as-grown BP films exhibit perfect orthogonally symmetric structure with high crystalline and purity, and also demonstrated with superior field effect and Hall mobility of over 1000 cm$^2$ V$^{-1}$ s$^{-1}$ and 1400 cm$^2$ V$^{-1}$ s$^{-1}$ at room temperature, respectively, and high current on/off ratio of 10$^6$, comparable to those exfoliated from BP bulk crystals. Superior infrared optical characteristics including strengthened infrared absorption and PL were explored and correlated with the lamellar structure of the BP films. Our newly developed gas-phase growth

approach therefore provides a new avenue for the scalable production of large-area, high-quality BP films, which promises their broad applications in versatile optoelectronic devices and atomically thin integrated circuits.

## Methods

**BP films growth**. Before BP growth, a layer of Au film (thickness $\leq 100$ nm) was deposited on silicon substrate by thermal evaporation deposition. Red P, Sn, SnI$_4$, and substrates were enclosed in evacuated silica glass tubes as the precursor for the growth. The tubes were sealed by a Partulab device (MRVS-1002). The details of the experiment were described in supporting information.

**Material characterizations**. The microstructure of the samples was characterized using a Quanta 400 FEG SEM, with the electron voltage minimized to 3 KV to avoid the damage of the BP samples. TEM image, SAED pattern, EDS spectrum, and elemental maps were obtained on a Titan 80–300 bright-field TEM with 100 KV acceleration voltage. XRD was used to study the crystal structure of the samples with a Bruker D8 diffractometer. XPS characterization was performed on an Axis Ultra DLD X-ray photoelectron spectroscopy. Raman spectra were also used to analyze the structure information of the as-obtained samples with a Lab-RAM HR evolution Raman spectrometer. The excitation was provided by visible laser light ($\lambda = 473$ nm) through a $\times 100$ objective. To avoid laser-induced damage of the samples, all spectra were recorded at low power levels ($P_{in} \sim 1.25 \, \mu W$).

**Preparation of cross-sectional TEM samples**. A dual beam instrument (FEI Nova NanoLab 600) was used for site specific preparation of cross-sectional samples. To protect the sample from ion beam bombardment, a 50 nm Au-Pd film was deposited on the whole sample thorough thermal evaporation. Then a 2 μm Pt strap sputtered on the surface at a chosen location, and Pt also providing mechanical stability to the cross-sectional slice after its removal. Trenches were milled around the Pt strap using a 5 KV Ga$^+$ beam with a current of 16 pA. The cross-sectional slice then be extracted and transferred to an copper half grid as required for TEM. A final gentle polish with Ga ions (at 2 kV and 8.7 pA) was used to remove side damage.

**Infrared spectroscopy measurements**. Infrared absorption spectroscopy was performed in the 400–4000 cm$^{-1}$ range using a Bruker Optics Fourier Transfer Infrared spectrometer (Vertex 70) integrated with a Hyperion 1000 microscope system. All the samples are prepared on the gold films, and the areas are larger than 2500 μm$^2$ in order to ensure the BP samples are larger than the laser diameter. The infrared PL measurement was excited by a focused laser of 800 nm (Tisapphire laser, 80 fs, 80 MHz). Infrared radiation (IR) light emission was collected by a reflective objective and fed into an infrared spectrometer (iHR320) equipped with an InSb IR detector. The signal was recorded by a computer. Usually, the sample was kept in the cryostat to realize the measurement at different temperature.

**Device fabrication and characterization**. Back-gated BP FETs were fabricated directly on the 285-nm-thick SiO$_2$ dielectric on degenerately doped silicon substrates. Optical microscopy was used to find atomically thin films (usually utilizing the $\sim 8$ nm thick) and atomic force microscopy (Vecco Dimension3100) was used to measure their thickness. EBL (JEOL JBX 5500) was used to define the source/drain patterns, and Au/Cr (90/10 nm)film was deposited using an Electron-beam evaporator (Ulvac Ei-5Z) followed by lift-off process to form the source-drain electrodes. To avoid oxidation of BP, the total time of exposure in ambient during fabrication was controlled within 1 h. All the transport property characterizations were carried out under vacuum ($10^{-4}$ Pa) inside a Lakeshore probe station with temperature from 10 K to 300 K. BP photodetectors reported here was designed as photo-transistor structure based on the same back-gated FETs. Hall measurements were carried out in four-probe systems (Oxford Instruments, Teslatron PT system), with magnets up to 12 T. The temperature ranged from 1.5 to 300 K. The magnetic field was applied perpendicular to the devices. Photoresponse was measured under the illumination with laser wavelengths from 980 nm to 4 μm, which was in the range from near to mid-infrared. The mid-infrared photoelectric responses of the devices were measured in the cryostat (ST-500, Janis) equipped with the electric measurement unit and CaF$_2$ optical window. After the cryostat being pumped to and maintained at $10^{-5}$ mbar, the electric or photoelectric signals of the devices was measured and recorded by a semiconductor characterization system (Keithley 4200). For the measurements of photoelectric response, a mid-infrared laser (wavelength can be tuned from 2600 to 4200 nm) was focused to the devices with the light spot diameter of 7 mm, using a CaF$_2$ lens.

## Data availability

The authors declare that the data supporting the findings of this study are available within the paper and its Supplementary Information files. The source data underlying Fig. 1d, f, g, 2f–i, 4a–f and supplementary Figs 1, 5a, b, 7c, 10, 11a, b, 12a, b, 13b–d, and 14a–d are provided as a source data.

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

## Acknowledgements

We thank Wenzhi Yu for finalizing the schematic images. This work was supported by the National Natural Science Foundation of China (grant nos. 61922082, 61875223, 51525202, U19A2090, 51802199, 61435010, 51772088, 61575089 and 11525415). The support from the Vacuum Interconnected Nanotech Workstation of Suzhou Institute of Nano-tech and Nano-bionics, Chinese Academy of Sciences is also acknowledged.

## Author contributions

K.Z., Y.X. conceived the original idea. K.Z., H.Z., and A.P. supervised the project. Y.X. and X.S. carried out the BP synthesis and structure characterizations. H.T.Z. and L.S. performed the bright-field-TEM imaging experiments, with assistance from Z.H. Y.X. and Z.Z. fabricated BP field-effect transistor devices and performed the electrical measurement. J.G. and X.X. conducted the Hall measurements. Q.Z. and Y.Z. performed the infrared PL and photodetecting measurements. All authors helped in analyzing and interpreting the data. Y.X., Q.Z., A.P., and K.Z. wrote the paper and all authors commented on it.

## Competing interests

The authors declare no competing interests.
