## [Peer Review File · Nature Communications]

Reviewers' comments:

Reviewer #1 (Remarks to the Author):

In this manuscript the authors demonstrate an approach of growing crystalline BP thin films on Si (or SiO₂/Si) substrate. Gold and tin were introduced to form Au₃SnP₇ as nucleation seeds at 750 °C for subsequent BP growth at 500 °C. The authors claim that high mobility and high on-off ratio transistors were achieved with the synthesized BP films. In addition, they observed enhanced infrared photoluminescence and extinction in the synthesized BP film.

Compared to existing works on BP growth, it seems at first sight that this work achieved an improvement in terms of mobility, transistor on-off ratio, BP crystallinity and lateral size. However, there are a few points that need further elaboration in order to justify this work as a real advancement for thin film BP growth.

1. The high on-off ratio of transistors can only be obtained with BP films of a small thickness (~8 nm). However, BP films with such thickness only achieved a lateral size ~30 μm by their growth method, offering hardly any advantages over BP flakes exfoliated from bulk crystal. This fact does not support their claim of large-scale growth which is suitable for compact integrated circuits.

2. The substrate was initially coated with a layer of Au (<= 100 nm). Is there any remaining Au (or Au₃SnP₇) on the substrate for the area that is not covered by BP? Does the remaining Au (or Au₃SnP₇) form continuous films or discrete islands? And what is the size of them? As the Au & Sn could become impurities/contamination on the chip, a more detailed reveal of their distribution and existing form could help to clear (or unfortunately verify) the scepticism from readers.

3. What's the selection criteria for the growth temperature and time? How does the temperature impact on the Au₃SnP₇ formation/nucleation/growing process and the quality of the BP film? Thermal budget is critical for material growth as it limits the integration compatibility and potential applications. Possibility to further decrease the thermal budget could make the growth approach more versatile and attractive. Including more analysis of the design criteria for the growth condition would help to evaluate the full potential of this method for BP growth.

4. What parameters determine the crystal orientation of the Au₃SnP₇ nucleation seeds? Are the crystal orientations the same for all the nucleation seeds on one substrate? The anisotropic in-plane lattice structure is a unique property of BP which is important to device performance, therefore, providing more information of crystal orientation and discussions on potential methods to control it could add more value to the manuscript.

5. Line 92-94: "However, the extremely high-pressure condition restricts the accessible growth substrates and development of complementary heterostructure devices based on BP."

(5-1) Why does high pressure limit the choice of substrates? Is it because of the mechanical robustness or some other reasons? What types of substrates do not stand high pressure? And why is this type of substrate important to the "development of complementary heterostructure devices based on BP"? A specific explanation would make a more convincing statement instead of a general excuse to depreciate other works.

(5-2) What was the pressure for BP growth in this work? The authors highlighted the pressure issue in the introduction yet didn't mention the value of this parameter in their work.

6. Is the substrate Si or SiO₂/Si (Si with oxide coating)?

In Fig. 1b & 1e & 3b, the substrate is labelled as "Si", while in Fig. 3a the Au₃SnP₇ is on a SiO₂ layer. The inconsistent labelling/description is quite confusing. Are there two types of substrate being used (Si and SiO₂/Si) or only SiO₂/Si? The authors should make it clear. If only SiO₂/Si substrates were used, then the "on silicon" description in the title is misleading.

Moreover, this growth method is not substrate-selective according to its nucleation mechanism, any substrate that could stand the thermal budget should work. It seems the authors tend to emphasize "Si substrate" for some reason. Maybe they think "growth on Si" would make the growth method CMOS compatible. However, their key step of introducing Au makes the method not compatible with CMOS technology anymore, and the thermal budget (1 hour @ 750 °C) is not compatible with back-end-of-line CMOS either.

7. The authors claim "strengthened infrared absorption" possibly due to the lamellar structure of the grown BP films.

(7-1) The extinction ($1-T/T_0$) does not always equal to absorption. The increased extinction could be a result of increased reflection caused by the lamellar structure.

(7-2) The infrared absorption difference between the grown BP film and exfoliated BP film may come from the additional Au₃SnP₇ layer underneath the grown films.

8. Line 117&120: the square symbol "□" should be "°C".

9. According the sequence & Fig. number mentioned in the text, Fig. f (currently the $1-T/T_0$) should be inserted before the current Fig. d.

Reviewer #2 (Remarks to the Author):

The authors report some interesting results on the growth of black phosphorus on a large scale. This is a long-standing problem and there are a few demonstrations as discussed in the manuscript but large-scale high-quality black phosphorus is still not available. This work is interesting and the authors show that the catalytic approach could also be used to grow large-scale black phosphorus on a wafer scale. I have to say that this long-standing problem still has not been resolved but the results are interesting and may provide a new perspective for future research. As a result, I do recommend its publication after addressing the following three issues.

(1) About the thickness of the grown black phosphorus. It seems that the grown black phosphorus is very thick (the thinnest is 5 nm). In many applications especially in electronics, thin black phosphorus is needed. How can they grow ultrathin black phosphorus?

(2) Gold is extensively used in growth. For silicon, the gold is the least desirable material because it can form deep centers, which can significantly modify the silicon properties. How will the authors resolve this gold contamination problem?

(3) The authors mentioned excitons in PL. In such thick black phosphorus, I think the excitonic effect can be completely ignored.

Reviewer #3 (Remarks to the Author):

This manuscript reports the direct epitaxial growth of large-scale highly crystalline BP films up to millimetre size on insulating Si substrates with lateral and thickness control. The as-grown BP films exhibit excellent electrical and optical properties, comparable to BP flakes exfoliated from bulk crystals grown by the mineraliser-assisted method. This work has the potential to be of great interest for the fast-growing field as till date, there aren't any reports for the successful wafer-scale growth of highly crystalline few-layer to monolayer BP on substrates. Unfortunately, the novelty of the paper is diminished as the growth control demonstrated is poor (e.g. authors report BP of size 20 μm with 100 nm thickness, 200 μm with 150 nm thickness, 4 mm with 10 μm thickness). Additionally, the quality of the thinnest BP ~ 5 nm is not characterized and evaluated. As mentioned by the authors in line 81, the growth of crystalline thin BP on substrates will be of

greater interest to the scientific community as exotic properties emerge when quantum confinement is present. Hence, the thin BP grown should be the main focus of the paper and more efforts should be made to increase the lateral size (reported $\sim 10 \mu\text{m}$ by the authors).

There are several claims that are not well-supported and should be addressed:

1. It is not clear in figure 1 and line 138 of the manuscript how the small BP nanosheets fuse to become a large crystalline BP film. It would be more convincing if the authors can characterize the lattice alignment of the BP nanosheets during the merging process. More specifically, if a highly crystalline BP were to result from the merging of two or more BP grains, the grains should be aligned and no grain boundaries should be present at the interface (Nature 570, pages 91–95, 2019). A careful study using polarized SHG, HRTEM or STM of the grains is needed here.
2. The authors briefly characterized the crystallinity of as-grown BP in figures 1g and 3e (mentioned in manuscript, line 218) using Raman spectroscopy and TEM (SAED), respectively. Raman spectroscopy and TEM are very local techniques which only probe a small selected area of the samples. LEED or ARPES measurements at several locations across the as-grown BP films should be provided to support such a major claim.
3. In line 272 of the manuscript, the authors mentioned that the BP films are stable due to the increase in PL intensity with increasing laser power. A surface morphological characterization such as AFM should be carried out before and after laser illumination to substantiate this claim as there are other possibilities explaining their observation.

Additionally, there are several less critical details that should be worked on by the authors:

4. There is insufficient experimental information to reproduce the growth of BP, in particular how the gold film was prepared.
5. The thickness of BP characterized should be more consistent, or even standardized, and properly mentioned throughout the manuscript and supporting information.
6. What is the experimental yield of BP? Elaborate lines 139-141 "...may be achieved..."
7. What is the elemental composition of the resulting BP on Si substrate? Are Au, I, and Sn still present and how much? How do the contaminants affect the electronic and optical properties of the as-grown BP if so?
8. In lines 126-127, the space group or symmetry group of Au_3SnP_7 and BP should be mentioned. The lattice parameters of BP should be explicitly mentioned as well.
9. Did the authors observe Raman peaks belonging to Au_3SnP_7 , since XRD peaks were observed, originating from Au_3SnP_7 below the BP film (line 154).
10. Line 161, the authors should do a proper fitting of the XPS peaks and provide a more accurate binding energy (BE). The reported BE is lower than the maxima of the XPS peak in figure S5.
11. Is there a carrier gas used for the vapor transport? The nature of the thermodynamic transport of P4 molecules in line 167 is not clear.
12. Line 202 "attend", please check for language use.
13. In fig 4a, is the reverse sweep shown? Is there any hysteresis?
14. The y-axis scale of fig 4d does not correspond to line 250 of the manuscript.
15. A reference should be added for line 265-266 "Usually, with the increasing of the thickness, the accumulated defects 266 in BP will lead to the broadening of the PL peaks."
16. A proper check for grammatical errors should be carried out.

Point-by-point response to the reviewer comments for manuscript entitled
“ Epitaxial nucleation and lateral growth of high-crystalline black phosphorus
films on silicon” (NCOMM S-19-29373)

Reviewer #1 (Remarks to the Author):

In this manuscript the authors demonstrate an approach of growing crystalline BP thin films on Si (or SiO₂/Si) substrate. Gold and tin were introduced to form Au₃SnP₇ as nucleation seeds at 750 °C for subsequent BP growth at 500 °C. The authors claim that high mobility and high on-off ratio transistors were achieved with the synthesized BP films. In addition, they observed enhanced infrared photoluminescence and extinction in the synthesized BP film.

Compared to existing works on BP growth, it seems at first sight that this work achieved an improvement in terms of mobility, transistor on-off ratio, BP crystallinity and lateral size. However, there are a few points that need further elaboration in order to justify this work as a real advancement for thin film BP growth.

We appreciate the reviewer for the valuable comments that “ there are a few points that need further elaboration in order to justify this work as a real advancement for thin film BP growth”, and would like to address the issues in the following:

1. The high on-off ratio of transistors can only be obtained with BP films of a small thickness (~8 nm). However, BP films with such thickness only achieved a lateral size ~30 μm by their growth method, offering hardly any advantages over BP flakes exfoliated from bulk crystal. This fact does not support their claim of large-scale growth which is suitable for compact integrated circuits.

Thanks for your kind comments.

Although the lateral size of the as-grown BP films with small thickness (thickness ~8 nm) is not large, the method of direct growth on substrate exhibits much superiority in applications compared with the mechanical exfoliation. Usually, transfer BP flakes exfoliated from bulk crystal needs scotch tape, which suffers the inevitable residues and the low yield. While, growing BP films directly on substrate could promise the cleanness of the BP films as well as the high efficiency with batches of growth. Furthermore, we may grow large-scale thicker BP films in high quality through proposed method, like sub-millimeter size when thickness around or over 100 nm, which is very promise for infrared optoelectronics. As previously reported, near-infrared BP photodetector for high-resolution imaging is achieved with 120 nm-thick multilayer BP (M. Engel et al., *Nano Letters* 14, 6414-6417, 2014). More excitingly, optimum device configuration of BP (~150 nm thick)/MoS₂ (15 nm thick) heterojunction were explored as mid-infrared polarization-resolved photodiodes with high detectivity at room temperature (J. Bullock et al., *Nature Photonics* 12, 601-607, 2018). This is a great advantage of the growth over exfoliation, where it is impossible to achieve such large-area BP films from bulk crystals.

And, we would never say that the growth at current stage is suitable for compact integrated circuits. What we claimed in this work, as shown in the manuscript title, is solving the nucleation problem and realizing lateral growth of high-crystalline BP films on dielectric substrates as silicon. In the manuscript, we also described in this way, as “an important step

forwards large-scale preparation of crystallized BP films and open the door for broad applications in scalable optoelectronic devices and compact integrated circuits” (line 16-17, Abstract) rather than “we can grow large-scale monolayer or few-layer BP films already for transistors and/or further compact integrated circuits”. Growth of high-quality BP on a large scale in a fine controllable way is a long-standing problem. Though it has not been resolved completely, we believe our current work moves forwards a critical step, and may provide new perspectives for future research.

2. The substrate was initially coated with a layer of Au (≤ 100 nm). Is there any remaining Au (or Au_3SnP_7) on the substrate for the area that is not covered by BP? Does the remaining Au (or Au_3SnP_7) form continuous films or discrete islands? And what is the size of them? As the Au & Sn could become impurities/contamination on the chip, a more detailed reveal of their distribution and existing form could help to clear (or unfortunately verify) the scepticism from readers.

We appreciate for the reviewer’s comment. During the growth process, Au film contracted into nano-islands and reacted with P and Sn to form Au_3SnP_7 on silicon during the heating process at 750°C . The formation of Au_3SnP_7 can be described by the following equations:

After reaction, Au and Sn formed as Au_3SnP_7 on the substrate and no pure element was left, therefore, there is no impact on the chip induced from Au and Sn.

We indeed observed Au_3SnP_7 islands remaining on the substrate after growth, distributed among the area that is not covered by BP. The lateral size of discrete Au_3SnP_7 islands ranging from hundred nanometers to several micrometers, as displayed in Figure 1 showing below. This eliminates the possibility of gold or Sn contaminant by introducing recombinant levels through Au-Si or Sn-Si.

Figure 1. The Au_3SnP_7 remaining on the substrate where the area that is not covered by BP.

3. What's the selection criteria for the growth temperature and time? How does the temperature impact on the Au_3SnP_7 formation/nucleation/growing process and the quality of the BP film? Thermal budget is critical for material growth as it limits the integration compatibility and potential applications. Possibility to further decrease the thermal budget could make the growth approach more versatile and attractive. Including more analysis of the design criteria for the growth condition would help to evaluate the full potential of this method for BP growth.

Thanks for your valuable comment. The temperature in the proposed growth method was carefully designed and optimized. In our experiments, red phosphorus was used as precursor. According to the phase diagram of phosphorus, the sublimation temperature of red phosphorus (RP) is 416 °C and the phase transition of P_4 to BP happens around 500 °C. First, we set the temperature at the heating stage somehow higher to be 750 °C. This is available for the complete RP sublimation into P_4 vapor, SnI_4 decomposed to Sn and I_2 , and also Au film contracted to be nano-islands. When cooling from 750 °C to 500 °C, Au reacted with P and Sn forming into distributed Au_3SnP_7 nano-islands on silicon, which plays the critical role of nucleation of BP. The fluctuation of the temperature of 750 °C at these stages would not affect a lot on the formation of Au_3SnP_7 and further growing process. One may decrease it to some lower temperature if expecting for decreasing the thermal budget. But the growth temperature of 500 °C is very important, which determines the success of the phase transition and quality of the BP films. Lower temperature will result in low-crystallinity of the film or even only final product of white P mixed with RP.

4. What parameters determine the crystal orientation of the Au_3SnP_7 nucleation seeds? Are the crystal orientations the same for all the nucleation seeds on one substrate? The anisotropic in-plane lattice structure is a unique property of BP which is important to device performance, therefore, providing more information of crystal orientation and discussions on potential

methods to control it could add more value to the manuscript.

Thanks for your kind suggestion.

Indeed, the anisotropic characteristics of BP, originated from its unique puckered lattice structure, is important and beneficial for various device applications, such as broadband and chirality-sensitive photodetection, catalysts, sensors, etc. Various anisotropic physical properties of BP such as optical, mechanical, thermoelectric, and electrical conductance etc. are related to their differences along zigzag and armchair directions. As clarified in the manuscript, the synthesized BP films are polycrystalline films with no preferred orientation. However, the as-grown BP films in our work exhibit the same anisotropic lattice structure and anisotropic property as the BP obtained from the mechanical exfoliation, which can be seen in Fig. 3e of the manuscript and Figure S12 in the Supporting Information.

We believe that growing large single-crystalline BP atomic layers and further controlling the crystal orientations is a very interesting and meaningful project. We would pay more effort on it in the future research, e.g. through controlling the crystal orientation of the Au_3SnP_7 nucleation seeds and exploring its relationship with the BP crystal orientation as suggested by the reviewer.

5. Line 92-94: “However, the extremely high-pressure condition restricts the accessible growth substrates and development of complementary heterostructure devices based on BP.”

(5-1) Why does high pressure limit the choice of substrates? Is it because of the mechanical robustness or some other reasons? What types of substrates do not stand high pressure? And why is this type of substrate important to the “development of complementary heterostructure devices based on BP”? A specific explanation would make a more convincing statement instead of a general excuse to depreciate other works.

Thanks for the reviewer’s kind comment. High-pressure condition is not compatible with CMOS technology, which will restrict its further development for complementary heterostructure devices. And, some fragile substrates that are important for optoelectronics like InP would not be applicable if with the extremely high-pressure growth condition. We agree with the reviewer that, the description in our manuscript regarding to the “extremely high-pressure condition restricts the accessible growth substrates and development of complementary heterostructure devices based on BP” is not proper. Therefore, we have corrected it as following, “However, the extremely high-pressure might be a disadvantage to the choice of the growth substrates for BP”, in manuscript page 3, line 18.

(5-2) What was the pressure for BP growth in this work? The authors highlighted the pressure issue in the introduction yet didn’t mention the value of this parameter in their work.

The pressure for BP growth in this work, with RP sealed in a vacuum tube, can be calculated according to Clausius-Clapeyron equation:

$$pV=nRT$$

where p is the pressure, V is the gas volume, n is the amount of precursors, R is the molar gas constant and T is the temperature. In our experiments, we sealed 50 mg red phosphorus in quartz tubes (Inner diameter=8mm, length=12 cm), so $n=1.61\times 10^{-3}$ mol and $V=6.03$ cm^3 , meanwhile, $R=8.314$ $\text{J}\cdot\text{mol}^{-1}\cdot\text{K}^{-1}$ and $T=773$ K. Accordingly, pressure is calculated to be about

1.7 MPa, which is much lower compared with the method of transforming RP to BP at 1.5 GPa and even 10 GPa as previously reported.

6. Is the substrate Si or SiO₂/Si (Si with oxide coating)?

In Fig. 1b & 1e & 3b, the substrate is labelled as “Si”, while in Fig. 3a the Au₃SnP₇ is on a SiO₂ layer. The inconsistent labelling/description is quite confusing. Are there two types of substrate being used (Si and SiO₂/Si) or only SiO₂/Si? The authors should make it clear. If only SiO₂/Si substrates were used, then the “on silicon” description in the title is misleading. Thanks for your careful review. Si covered with SiO₂ is widely used as the substrate in electrical and optoelectronic devices. In our work, the BP film is grown on SiO₂/Si substrates. In order to avoid the misleading, we have corrected the inconsistent labeling/description in the manuscript, including Figure 1b, 1e, Figure 3b, and as displayed in the following.

Fig. 1 in the manuscript.

Fig. 3 in the manuscript.

Moreover, this growth method is not substrate-selective according to its nucleation mechanism, any substrate that could stand the thermal budget should work. It seems the authors tend to emphasize “Si substrate” for some reason. Maybe they think “growth on Si” would make the growth method CMOS compatible. However, their key step of introducing Au makes the method not compatible with CMOS technology anymore, and the thermal budget (1 hour @ 750 °C) is not compatible with back-end-of-line CMOS either.

We thanks for the reviewer’s valuable comment. During the growth process, Au film contracted into nano-islands and reacted with P and Sn to form Au_3SnP_7 , and hence there is no Au residues/contaminants left on the silicon substrate. As clarified above in question 3, the

heating temperature of 750 °C mainly for the RP sublimation and SnI₄ decomposition may be decreased to some lower temperature if expecting for decreasing the thermal budget. And, the subsequent device fabrication processes including lithography, electrode deposition, etc. are all carried out at CMOS compatible conditions. The proposed method of growing BP directly on dielectric substrates including Silicon, plus these accessibilities, provides possibilities for the development of CMOS compatible BP based optoelectronics.

7. The authors claim “strengthened infrared absorption” possibly due to the lamellar structure of the grown BP films.

(7-1) The extinction ($1-T/T_0$) does not always equal to absorption. The increased extinction could be a result of increased reflection caused by the lamellar structure.

We appreciate the reviewer for the valuable suggestion. Indeed, the extinction includes absorption and reflection. The periodic structure of the material may result Bragg reflection effect and cause the increase of reflection. However, the gap distance and the thickness of the unit cells of the BP film are not extremely regular, which doesn't satisfy the demand of Bragg reflection. Therefore, we think the increase of the extinction is most probably related to the absorption rather than reflection.

(7-2) The infrared absorption difference between the grown BP film and exfoliated BP film may come from the additional Au₃SnP₇ layer underneath the grown films.

The infrared absorption measurements were carried out with BP films transferred on silicon substrates coated with Au films. This will eliminate any background problem from the substrates during the infrared spectroscopy characterizations. Accordingly, in the sample preparations, BP films in the same 120 nm thick were exfoliated from as-grown thicker BP films (like several microns in thickness as shown in Figure S6 and S8) and conventional bulk BP crystals onto Au/Si substrates in a similar way. Therefore, there will be no impact from Au₃SnP₇ layer on infrared absorption.

8. Line 117&120: the square symbol “□” should be “°C”.

Thanks for the reviewer's careful review. We have revised them accordingly. And, in the revised manuscript, we have checked all the typos thoroughly.

9. According the sequence & Fig. number mentioned in the text, Fig. f (currently the $1-T/T_0$) should be inserted before the current Fig. d.

We appreciate for the reviewer's careful review. Fig. d to f were disordered, Fig. f should be Fig. d, Fig. d should be Fig. e, Fig. f should be Fig. f. The order of the figure has been corrected, and exhibited as the following.

Fig. 4 in the manuscript

Reviewer #2 (Remarks to the Author):

The authors report some interesting results on the growth of black phosphorus on a large scale. This is a long-standing problem and there are a few demonstrations as discussed in the manuscript but large-scale high-quality black phosphorus is still not available. This work is interesting and the authors show that the catalytic approach could also be used to grow large-scale black phosphorus on a wafer scale. I have to say that this long-standing problem still has not been resolved but the results are interesting and may provide a new perspective for future research. As a result, I do recommend its publication after addressing the following three issues.

We appreciate the reviewer for the positive comment to our work as “This work is interesting.....and may provide a new perspective for future research. As a result, I do

recommend its publication after addressing the following three issues”, and would like to address the issues one by one in the following:

1. About the thickness of the grown black phosphorus. It seems that the grown black phosphorus is very thick (the thinnest is 5 nm). In many applications especially in electronics, thin black phosphorus is needed. How can they grow ultrathin black phosphorus?

We thank for the reviewer’s valuable comment. Controlling the thickness of the BP film is a tough task. In our work, several growth routes for reducing the P_4 source involved in the transformation into BP, including inverse thermal transportation, confining the diffusion of P_4 with limited space, etc., were employed to control the thickness of the BP film. Based on our growth strategy, BP films in various thicknesses that as thin as few-layer BP in 5 nm can be obtained. However, it is still difficult to grow thinner BP film or even monolayer.

Thinner BP as monolayer, bilayer BP films may be expected for higher integrations in complementary electronics. But we would like to say, thin BP films with thickness as 5-10 nm have already show very promising electrical properties (Nat. Nanotechnol. 9, 372-378, 2014). As reported, BP film with thickness around 10 nm exhibits carrier mobility as high as $1000 \text{ cm}^2\text{V}^{-1}\text{s}^{-1}$ as well as good on/off current ratio on the order of 10^5 , and regarded as a good choice for many electronic applications, such as FETs, photodetector, sensor, etc.

We agree with the reviewer that growing ultrathin BP would be very interesting and significant. Currently, we are trying to move forwards growing thinner BP films through optimizing growth set up together with parameters. We’re expecting for new progress in growing thinner or even monolayer BP films in the near future.

2. Gold is extensively used in growth. For silicon, the gold is the least desirable material because it can form deep centers, which can significantly modify the silicon properties. How will the authors resolve this gold contamination problem?

Thanks for your valuable comment. During the growth process, Au film contracted into nano-islands and reacted with P and Sn to form Au_3SnP_7 , with no pure Au residues left on the silicon substrate (as demonstrated in Fig. 1 and Fig. 3 of the manuscript and Fig. S2 in the supporting information). This would eliminate the possibility of gold contaminant by introducing recombinant levels through Au-Si.

3. The authors mentioned excitons in PL. In such thick black phosphorus, I think the excitonic effect can be completely ignored.

We appreciate the reviewer for the comment. If exciton binding energy lower than the $k_B T$ (where k_B is the Boltzmann constant, T is the temperature), the exciton will dissociate due to the thermal activation. In our work, the PL was measured at 77 K, and the $k_B T$ is 6.6 meV. The exciton binding energy of BP with 200 nm thickness is about 16 meV (J. Phys. C: Solid State Phys. 17, 1839, 1984), which is higher than the $k_B T$ at 77 K. Therefore, the exciton can exist at that circumstance and the excitonic effect should be considered.

Reviewer #3 (Remarks to the Author):

This manuscript reports the direct epitaxial growth of large-scale highly crystalline BP films up to millimetre size on insulating Si substrates with lateral and thickness control. The as-grown BP films exhibit excellent electrical and optical properties, comparable to BP flakes exfoliated from bulk crystals grown by the mineraliser-assisted method. This work has the potential to be of great interest for the fast-growing field as till date, there aren't any reports for the successful wafer-scale growth of highly crystalline few-layer to monolayer BP on substrates. Unfortunately, the novelty of the paper is diminished as the growth control demonstrated is poor (e.g. authors report BP of size 20 μm with 100 nm thickness, 200 μm with 150 nm thickness, 4 mm with 10 μm thickness). Additionally, the quality of the thinnest BP ~ 5 nm is not characterized and evaluated. As mentioned by the authors in line 81, the growth of crystalline thin BP on substrates will be of greater interest to the scientific community as exotic properties emerge when quantum confinement is present. Hence, the thin BP grown should be the main focus of the paper and more efforts should be made to increase the lateral size (reported ~ 10 μm by the authors).

Thanks for the reviewer's kind comments as "This work has the potential to be of great interest for the fast-growing field as till date, there aren't any reports for the successful wafer-scale growth of highly crystalline few-layer to monolayer BP on substrates". And, the queries and suggestions raised here are available for us to improve the manuscript.

Before addressing the issues one-by-one, we would like to clarify that both atomically thin BP films like monolayer BP and multilayer BP films in thickness of several to hundreds of nanometers are of great interest to the scientific community. There would be exotic properties accompanied with the quantum confinement of the atomically thin BP films. On the other hand, multilayer BP films also show great potential for the development of novel devices.

As reported, BP film with thickness around 5-10 nm exhibits carrier mobility as high as 1000 $\text{cm}^2\text{V}^{-1}\text{s}^{-1}$ as well as good on/off current ratio on the order of 10^5 , and regarded as a good choice for many electronic applications, such as FETs, photodetector, sensor, etc. (Nat. Nanotechnol. 9, 372-378, 2014). And also, thicker BP films like those with thickness around or over 100 nm, are very promise for infrared optoelectronics. E.g., near-infrared BP photodetector for high-resolution imaging is achieved with 120 nm-thick multilayer BP (M. Engel et al., Nano Letters 14, 6414-6417, 2014). More excitingly, optimum device configuration of BP (~ 150 nm thick)/MoS₂ (15 nm thick) heterojunction were explored as mid-infrared polarization-resolved photodiodes with high detectivity at room temperature (J. Bullock et al., Nature Photonics 12, 601-607, 2018).

As introduced in the manuscript, there has been no report on the success of growing crystalline BP thin films directly on substrates. Hence, the main focus of the paper, as we claimed in the manuscript title, is solving the nucleation problem and realizing lateral growth of high-crystalline BP films on dielectric substrates as silicon. And now, we have realized the growth of BP films with thickness control from 5 nm to microns. Lateral sizes of ~ 30 for 10 nm thick BP films, over 200 μm for 100 nm thick ones and 4 mm for 10 μm thick films have been achieved. Growth of high-quality BP on a large scale in a fine controllable way is a long-standing problem. Though it has not been resolved completely, we believe our current work moves forwards a critical step, and may provide new perspectives for future research.

We agree with the reviewer that more efforts should be made to increase the lateral size of

ultrathin BP films. It is equally important with some other issues remaining for BP thin films growth, like growing large single-crystalline BP atomic layers, controlling the crystal orientations, etc. We would pay more efforts on them in the future research.

In the following, we would like to try to address the issues in detail:

There are several claims that are not well-supported and should be addressed:

1. It is not clear in figure 1 and line 138 of the manuscript how the small BP nanosheets fuse to become a large crystalline BP film. It would be more convincing if the authors can characterize the lattice alignment of the BP nanosheets during the merging process. More specifically, if a highly crystalline BP were to result from the merging of two or more BP grains, the grains should be aligned and no grain boundaries should be present at the interface (Nature 570, pages91–95, 2019). A careful study using polarized SHG, HRTEM or STM of the grains is needed here.

We appreciate the reviewer's valuable suggestions. BP nanosheets epitaxially grow on the Au_3SnP_7 nuclei center, then thermodynamically grow and fuse across with each other to be large planar ones during the further heating process. To study the grain boundaries and grain size of the as grown BP film, dark filed TEM measurement was carried out as suggested (see Figure 2 as below). Diffraction patterns taken with larger aperture filters show the polycrystalline nature of the as-grown BP nanosheets with several micro meters in lateral size (Figure 2b). To determine the grain size of the synthesized BP film, diffraction-filtered imaging technique was employed. The resulting image reveals an intricate patchwork of grains connected by tilt boundaries, as depicted in Figure 2c, where adjacent grains were rendered into different colours corresponding to the different diffraction sets circled in Figure 2b. The rendered grain map indicates the grain size of the as-grown BP nanosheet is in a hundred of nanometers.

Figure 2. a, A typical TEM micrograph of the as-grown BP film. b, Diffraction pattern taken from a BP nanoflake with a larger aperture filter, reveals the polycrystalline structure of the sample. c, Corresponding diffraction-filtered image. The image reveals an intricate patchwork of grains that were rendered into different colours corresponding to the different diffraction sets circled in Figure 3f. The grain size of the as-grown BP nanosheet is examined to be a hundred of nanometers.

2. The authors briefly characterized the crystallinity of as-grown BP in figures 1g and 3e (mentioned in manuscript, line 218) using Raman spectroscopy and TEM (SAED),

respectively. Raman spectroscopy and TEM are very local techniques which only probe a small selected area of the samples. LEED or ARPES measurements at several locations across the as-grown BP films should be provided to support such a major claim.

We appreciate the review's valuable suggestion. As clarified above in question 1, it has been demonstrated with the dark field TEM measurement that the as-grown BP film in our work is polycrystalline constructed by single crystal grains in a hundred of nanometers. To study the crystallinity of the BP film in larger scale, diffraction patterns were taken at different locations of the film with larger selected-area diffraction aperture of 700 nm in diameter (Fig. S9a, insert). The sharp and intense diffraction spots in the typical SAED pattern (shown in insert of Fig. S9b) proves the high crystallinity nature of the BP film over hundreds of nanometers.

We have added the Figure S9 and corresponding description into the Supporting Information of the manuscript.

Figure S9 | a, A typical TEM micrograph of as-grown BP film. The inset shows the size of the aperture filter. b, displays a representative SAED pattern of the BP film with the larger selected-area diffraction aperture, demonstrating its single-crystalline characteristic.

3. In line 272 of the manuscript, the authors mentioned that the BP films are stable due to the increase in PL intensity with increasing laser power. A surface morphological characterization such as AFM should be carried out before and after laser illumination to substantiate this claim as there are other possibilities explaining their observation.

Thanks for your valuable suggestions. We carried out the AFM measurement to investigate the effect of the laser illumination on the BP film before and after PL. Before laser illumination, the sample exhibits a very smooth surface with almost no cracks or bubbles (as shown in Fig. 3a). After illuminated with laser power of 30 mW in a vacuum cryostat, the surface of the BP film displays still a flat and clean surface, shown in Fig. 3b. This result confirms the BP films are stable under laser illumination.

Figure 3. AFM image of the BP film before and after PL measurement. After illuminated with laser power of 30 mW in a vacuum chamber, the surface of the BP film displays still a flat and clean surface.

Additionally, there are several less critical details that should be worked on by the authors:

4. There is insufficient experimental information to reproduce the growth of BP, in particularly how the gold film was prepared.

Thanks for your comment. The gold film was deposited on substrate by thermal evaporation deposition, as we described in manuscript, method, BP film growth: “Before BP growth, a layer of Au film (thickness ≤ 100 nm) was deposited on Si substrate by thermal evaporation deposition.”

5. The thickness of BP characterized should be more consistent, or even standardized, and properly mentioned throughout the manuscript and supporting information.

Thanks for the kind suggestion. To facilitate different characterizations, as-grown BP films in appropriate thicknesses were utilized. For example, thinner films are available for gate modulations in electrical measurements. Hence, we fabricate FETs with BP films in ~ 5 - 10 nm thick for transport, hall measurements, etc. While infrared spectroscopy measurements prefer thicker films with strong absorption and excitation. For this, BP films with thicknesses of 50 nm to 200 nm were utilized in FTIR and infrared PL characterizations. And, the thickness of the BP has been mentioned throughout the manuscript and supporting information.

6. What is the experimental yield of BP? Elaborate lines 139-141 “...may be achieved...”

Usually, about 5 pieces of BP film with more than 100 μm in lateral size and 150 nm or less in thickness can be grow on each substrate, Figure 4a to c showing below display three typical samples at discrete locations on one substrate. The yield of thicker BP films is much higher, that is ~ 10 pieces of BP flakes with lateral size of millimeters and thickness of 200 nm or above can be obtained on each substrate, as shown in Figure 4d.

Figure 4. Representative optical images of the BP films grown on the Si/SiO₂ substrate. Figure a to c present 3 pieces of BP film at different locations of one substrate with more than 100 μm in lateral size and around 150 nm or less in thickness. Figure d displays large-area while thicker BP films grown on one substrate that was taken under low magnification of 10 × microscope. We can see ~10 pieces of BP flakes with lateral size of millimeters and thickness of 200 nm or above with this sample.

7. What is the elemental composition of the resulting BP on Si substrate? Are Au, I, and Sn still present and how much? How do the contaminants affect the electronic and optical properties of the as-grown BP if so?

Thanks for your valuable comment. The cross-sectional TEM image of the as-grown BP film on silicon substrate and the corresponding element mapping was presented in Fig. 3a. in the manuscript, where the sharp interfaces between Si, Au₃SnP₇ nucleation seed layer and BP film can be clearly observed. The element mapping images clearly display the purity of the as-grown BP film without any Au, Sn and I diffusion and contamination. The purity of the BP films was also supported with XPS characterizations as in Figure S5 in the Supporting Information. The purity and high crystallinity guarantees the excellent electrical and optical properties of the as-grown BP, for example, superior field effect and Hall mobility of over 1000 cm²V⁻¹s⁻¹ and 1400 cm²V⁻¹s⁻¹ at room temperature, respectively, and high current on/off ratio of 10⁶.

Fig 3a. in the manuscript. Cross-sectional TEM image of a BP film grown on silicon substrate and the corresponding element mapping of the P, Au, Sn, Si and O.

8. In lines 126-127, the space group or symmetry group of Au_3SnP_7 and BP should be mentioned. The lattice parameters of BP should be explicitly mentioned as well.

Thanks for your kind suggestion. The space group of the Au_3SnP_7 and BP is $P2_1/m$ (11) and $Cmce$ (64), respectively. And, we conducted further HRTEM characterizations to indicate the lattice parameters of Au_3SnP_7 and BP, as exhibited in the revised Fig. S4. As can be seen in Fig. S4a, the lattice spacing of the Au_3SnP_7 along (010) and (110) directions are 3.3 Å and 5.5 Å, respectively. While, the lattice parameters of BP along (100) and (001) directions are 3.3 Å and 4.4 Å, respectively (Fig. S4b). This characteristic makes Au_3SnP_7 served as a proper nucleation seed for the epitaxial growth of BP.

Fig. S4. a, HRTEM image of the Au_3SnP_7 nucleation seed grown on Si/SiO_2 substrate. Inset shows the corresponding SAED pattern of the Au_3SnP_7 , indicates its single-crystalline characteristic. b, HRTEM image of the BP film grown on Au_3SnP_7 nucleation seed, inset shows the corresponding SAED pattern of the BP film. c, Schematic image of the crystal structure of BP (top part) and Au_3SnP_7 (bottom part). It can be observed obviously BP along the (100) plane and Au_3SnP_7 along the (010) plane share very similar crystal structure to each other, which makes the P_4 phase prefer to transform to be BP and epitaxially nucleated on Au_3SnP_7 .

9. Did the authors observe Raman peaks belonging to Au_3SnP_7 , since XRD peaks were observed, originating from Au_3SnP_7 below the BP film (line 154).

We didn't observe any Raman peak belonging to Au_3SnP_7 accompanied with BP. XRD scans over the entire sample surface, while Raman measurement on very limit locations. Raman spectroscopy used in our experiment is equipped with a 100×objective. During the Raman

measurement, the laser spot focused on the surface of the BP in diameter of microns. It is possibly we either miss the area with Au_3SnP_7 nano-islands below the BP film or ignore any weak signal from it.

10. Line 161, the authors should do a proper fitting of the XPS peaks and provide a more accurate binding energy (BE). The reported BE is lower than the maxima of the XPS peak in figure S5.

Thanks for the reviewer's valuable suggestions. The XPS peak was fitted and two peaks $2p_{3/2}$ and $2p_{1/2}$ can be observed. The binding energy (BE) of the $2p_{3/2}$ and $2p_{1/2}$ peaks reported in the manuscript is lower than that in Figure S5b. Actually, the BE of the $2p_{3/2}$ and $2p_{1/2}$ peaks is 130.15 and 130.9 eV, respectively. We have corrected it in the manuscript, in page5, line 27.

Figure S5b. XPS spectra of the BP films grown on Si substrate

11. Is there a carrier gas used for the vapor transport? The nature of the thermodynamic transport of P_4 molecules in line 167 is not clear.

Thanks for your comment. There is no carrier gas for the vapor transport here. The BP film was synthesized by the mineralizer-assisted gas-phase transformation method, where precursors were sublimated into vapor and diffused thermodynamically towards the growth end. Therefore, the vapor transport of the P_4 molecules belongs to thermodynamical transportation.

12. Line 202 “attend”, please check for language use.

Thanks for your careful review. The word “attend” in Line 202 should be “tend” and we have corrected it in the manuscript.

13. In fig 4a, is the reverse sweep shown? Is there any hysteresis?

The reverse sweep of the FET with as-grown BP film is shown in Figure 5 as below, where a

hysteresis transfer curve can be observed. The hysteresis is a universal phenomenon in FETs which was probably due to the residues during the device fabrication, the charged traps in SiO_2 , the H_2O and O_2 from the ambient environment, etc.

Figure 5. Transfer curves of the as-grown BP film based FETs with forward and reverse sweep.

14. The y-axis scale of fig 4d does not correspond to line 250 of the manuscript. Thanks for the reviewer's careful review. Fig. d to f were disordered, where Fig. f should be Fig. d, Fig. d should be Fig. e, Fig. f should be Fig. f. The order of the figure has been corrected, which is shown as the following.

Fig. 4 in the manuscript

15. A reference should be added for line 265-266 “Usually, with the increasing of the thickness, the accumulated defects in BP will lead to the broadening of the PL peaks.”

Thanks for your comment. Corresponding references have been added as reference 28 and 29 in the proper place in the manuscript:

1. Study of defects and strain relaxation in GaAs/In_xGa_{1-x}As/GaAs heterostructures using photoluminescence, positron annihilation, and x-ray diffraction. J. Appl. Phys. 87, 8444-8450, (2000)

2. Size and defect related broadening of photoluminescence spectra in ZnO:Si nanocomposite films. Mater. Res. Bull. 47, 901–906, (2012)

These papers are added as reference 28 and 29 in the proper place in manuscript.

16. A proper check for grammatical errors should be carried out.

Thanks for your kind suggestions. The grammatical errors have been checked through the manuscript.

Reviewers' comments:

Reviewer #1 (Remarks to the Author):

The authors have addressed my questions, and I am satisfied with the revision made to the manuscript. I recommend its publication in Nature Communications.

Reviewer #2 (Remarks to the Author):

The authors tried to address my concerns in the revised version. However, I feel that they should improve their understanding. For example, they argue that in PL excitons should be considered because the binding energy is greater than kT . In principle, this statement is not wrong by itself, but the key here is which oscillator strength is larger, excitonic emission or bandgap emission? If excitonic emission dominates, then the PL is excitonic PL otherwise it is mainly due to bandgap emission. They need to provide evidence to support their claims.

Another example is Au contamination. The fact that you did not observe Au in SEM or optical micrographs does not mean there is no contamination. This evidence is too weak to draw any conclusion. Much improved characterization techniques are needed to measure the concentration of the contaminants. I can also find many other sentences and arguments which are not very convincing.

In short, overall this work is very good and would like to support its publication. However they need to go through the manuscript and check their logic and physics carefully.

Reviewer #3 (Remarks to the Author):

The authors have adequately replied to the issues outlined in the previous review, especially the unsupported claims. More specifically, there is now sufficient evidence to support the claim that the as-grown BP is highly crystalline, up to the order of hundreds of nanometers. The overall BP film however is polycrystalline. Additionally, stability of the as-grown BP had also been more thoroughly investigated.

The ability to grow substantially crystalline BP on substrates up to millimetre size is unprecedented. Interestingly, there is still much room for improvement regarding the large-scale crystallinity of the as-grown BP film, however, the authors have demonstrated that the synthesized BP films achieved remarkable electrical performance, comparable to exfoliated BP flakes. This paper poses an interesting question as to whether the bottom up approach to synthesize BP has the potential to surpass the top down approach in terms of electrical properties since much efforts have been dedicated to the study of exfoliated BP flakes but the mobility values reported are still rather far off the theoretical values. Additionally, this paper would serve as a rather high benchmark for the improvement in electrical and optical performance of synthesized BP. As this work demonstrates a huge advancement in BP growth, I would recommend it for publication in Nature Communications.

Point-by-point response to the reviewer comments for manuscript entitled
“Epitaxial nucleation and lateral growth of high-crystalline black phosphorus
films on silicon” (NCOMMS-19-29373A)

Reviewer #1 (Remarks to the Author):

The authors have addressed my questions, and I am satisfied with the revision made to the manuscript. I recommend its publication in Nature Communications.

We appreciate the reviewer for the positive comments that “I recommend its publication in Nature Communications”.

Reviewer #2 (Remarks to the Author):

The authors tried to address my concerns in the revised version. However, I feel that they should improve their understanding. For example, they argue that in PL excitons should be considered because the binding energy is greater than kT . In principle, this statement is not wrong by itself, but the key here is which oscillator strength is larger, excitonic emission or bandgap emission? If excitonic emission dominates, then the PL is excitonic PL otherwise it is mainly due to bandgap emission. They need to provide evidence to support their claims.

Another example is Au contamination. The fact that you did not observe Au in SEM or optical micrographs does not mean there is no contamination. This evidence is too weak to draw any conclusion. Much improved characterization techniques are needed to measure the concentration of the contaminants. I can also find many other sentences and arguments which are not very convincing.

In short, overall this work is very good and would like to support its publication. However they need to go through the manuscript and check their logic and physics carefully.

We appreciate the reviewer for the kind comments that “In short, overall this work is very good and would like to support its publication”. The suggestions raised are valuable for well improving the physics and logic of the work, and we would like to address the issues with corresponding revisions in the following:

We have checked the excitonic effect in BP carefully, and fully agree with the reviewer’s comment that “In such thick black phosphorus, the excitonic effect can be completely ignored”. In monolayer BP, the exciton binding energy is 0.3 eV, which is comparable to that of the bandgap energy, therefore, the excitonic effect should be considered. However, the exciton binding energy in our sample for PL (200 nm thick) is 16 meV, which is much lower than the bandgap energy (300 meV). Therefore, the PL here is dominated by bandgap emission. And, we have deleted the statement of “as well as the stability of the excitons that cause the narrower PL.” in the manuscript. The narrow of the PL of the lamellar BP is a very complicate process that is far from any conclusion in the current stage. As the key point of this work is the growth of high-crystalline BP film, we would try to investigate this PL phenomenon in the future study.

For the query about the possible Au contamination to the silicon substrate, we carried out additional SIMS (Secondary Ion Mass Spectrometry) and XPS (X-Ray Photoelectron Spectroscopy) measurements carefully to check it. First, SIMS was used to qualitatively

measure the diffuse depth of the Au in the SiO₂/Si substrate, as can be seen in the Fig. a. The normalized intensity of the Au drops dramatically to be vanishing at the depth of around 130 nm, which means the diffuse depth of the Au in SiO₂ is at most to be about 130 nm. As the thickness of the SiO₂ is 285 nm, therefore, we can conclude that the Au does not diffuse into Si beneath the SiO₂. Fig. b presents the 3D image of the Au (red part) diffusion in the SiO₂ (blue part), almost no Au can be observed in the substrate deeper than 130 nm. Moreover, XPS was used to quantitatively evaluate the concentration of the Au in the SiO₂. Fig. c displays the XPS spectrum of the substrate with the depth around 16 nm. Sharp O 1s and Si 2p peaks belonging to the SiO₂ present compared to the very weak Au 4f peak around 88 eV. The calculated percentage of the Au in the SiO₂ was exhibited in the Fig. d, which is around 0.01%. As reported before (phys. stat. sol. 222, 319, (2000)), the metal concentration about 10¹² cm⁻³ in Si substrate will degrade the performance of the substrate. In our experiment, the calculated Au concentration in the SiO₂ is about 2.8* 10⁵ cm⁻³, which is negligible that would not cause any impact to the substrate. We think the formation of the Au₃SnP₇ plays a critical role in inhibiting the diffusion of the Au into the SiO₂/Si substrate.

Fig. a and b, SIMS measurement of the Au distribution in the SiO₂/Si substrate. c, XPS measurement of the SiO₂/Si substrate, O 1s, Si 2p and Au 4f peaks were detected. d, The calculated percentage of the Au atom diffused in the SiO₂.

And, we have checked through the manuscript carefully as suggested and corrected some mistakes. For example, the description of the “Dirac point locates at 30V” is not correct, it is the **threshold gate voltage** rather than the Dirac point. We have corrected in the manuscript, page 8, paragraph 1, line 6.

Reviewer #3 (Remarks to the Author):

The authors have adequately replied to the issues outlined in the previous review, especially the unsupported claims. More specifically, there is now sufficient evidence to support the claim that the as-grown BP is highly crystalline, up to the order of hundreds of nanometers. The overall BP film however is polycrystalline. Additionally, stability of the as-grown BP had also been more thoroughly investigated.

The ability to grow substantially crystalline BP on substrates up to millimetre size is unprecedented. Interestingly, there is still much room for improvement regarding the large-scale crystallinity of the as-grown BP film, however, the authors have demonstrated that the synthesized BP films achieved remarkable electrical performance, comparable to exfoliated BP flakes. This paper poses an interesting question as to whether the bottom up approach to synthesize BP has the potential to surpass the top down approach in terms of electrical properties since much efforts have been dedicated to the study of exfoliated BP flakes but the mobility values reported are still rather far off the theoretical values. Additionally, this paper would serve as a rather high benchmark for the improvement in electrical and optical performance of synthesized BP. As this work demonstrates a huge advancement in BP growth, I would recommend it for publication in Nature Communications. We appreciate the reviewer for the positive comments that “As this work demonstrates a huge advancement in BP growth, I would recommend it for publication in Nature Communications”.

REVIEWERS' COMMENTS:

Reviewer #2 (Remarks to the Author):

The authors addressed my concerns and now it can be published.